# Effect of a Novel Flame Retardant on the Mechanical, Thermal and Combustion Properties of Poly(Lactic Acid)

**DOI:** 10.3390/polym12102407

**Published:** 2020-10-19

**Authors:** Mingjun Niu, Zhongzhou Zhang, Zizhen Wei, Wanjie Wang

**Affiliations:** College of Materials Science and Engineering, Zhengzhou University, Zhengzhou 450001, China; niumj@zzu.edu.cn (M.N.); zhangzhongzhou2010@163.com (Z.Z.); weizizhen0101@163.com (Z.W.)

**Keywords:** flame retardant, poly(lactic acid), mechanical properties, blends

## Abstract

Poly(lactic) acid (PLA) is one of the most promising biobased materials, but its inherent flammability limits its applications. A novel flame retardant hexa-(DOPO-hydroxymethylphenoxy-dihydroxybiphenyl)-cyclotriphosphazene (HABP-DOPO) for PLA was prepared by bonding 9,10-dihydro-9-oxy-10-phosphaphenanthrene-10-oxide (DOPO) to cyclotriphosphazene. The morphologies, mechanical properties, thermal stability and burning behaviors of PLA/HABP-DOPO blends were investigated using a scanning electron microscope (SEM), a universal mechanical testing machine, thermogravimetric analysis (TGA), differential scanning calorimetry (DSC), limiting oxygen index (LOI), vertical burning (UL-94) and a cone calorimeter test (CCT). The LOI value reached 28.5% and UL-94 could pass V-0 for the PLA blend containing 25 wt% HABP-DOPO. A significant improvement in fire retardant performance was observed for PLA/HABP-DOPO blends. PLA/HABP-DOPO blends exhibited balanced mechanical properties. The flame retardant mechanism of PLA/HABP-DOPO blends was evaluated.

## 1. Introduction

Plastics originated from petrochemicals cause serious environmental problems due to their non-degradable nature. Biodegradable plastics are a promising replacement for conventional synthetic plastics. Poly(lactic acid) (PLA) is an attractive biopolymer for packaging engineering [1], biomedical fields [2], automobile engineering [3,4] and the electricity industry [5] because of its biodegradability, non-toxicity, high optical transparency and good mechanical, thermal and electrical properties [6,7]. However, PLA has some shortcomings such as high flammability and poor toughness, which can easily lead to fire accidents and limit its further applications. Therefore, investigations into PLA for its excellent thermal, mechanical and flame retardant properties have attracted more and more attention [8,9].

In recent years, several approaches have been applied to improve the flame retardant properties of PLA [10,11,12]. Non-halogen flame retardants such as phosphorus-containing silsesquioxane [13,14,15], intumescent flame retardants [16,17,18,19,20] and nitrogen phosphorus flame retardants [21,22,23] were employed to overcome this drawback. For example, Chen et al. [24] evaluated the efficiency of a novel flame retardant heax-(*N*,*N*’,*N*’’-tris-(2-amino-ethyl)-(1,3,5)triazine-2,4,6-triamine)cyclotriphosphazene (HTTCP) for PLA. They found that when the concentration of HTTCP was 25 wt%, the blends showed a 25.2% value of limiting oxygen index (LOI) and a lower peak of heat release rate (pk-HRR) compared with neat PLA under the same measurement conditions. Xi et al. [22] characterized flame retardancy, using a cone calorimeter test, and dripping properties of PLA/10 wt% TGIC-DOPO blend. The results showed that the flame retardant promoted breaking up and dripping early, released fragments with quenching and dilution effects, and improved the flame retardancy of PLA. Gu et al. [23] synthesized multi-walled carbon nanotubes (MWCNT)-DOPO-OH with a core-shell nanostructure by a three-step process. MWCNT-DOPO-OH was introduced into aluminum hypophosphite (AHP)/PLA flame retardant systems via melt blending to improve both flame retardancy and mechanical properties. Apart from types of organic hydroxide multi-walled carbon nanotubes and sepiolite nanoclay, biofibers [25,26] can also greatly elevate LOI, dripping, vertical burning (UL-94) and thermal properties of PLA blends. Although many types of flame retardants can improve effectively the flame retardancy of PLA, the mechanical properties of PLA decrease greatly due to high loading of flame retardants.

Recently, a novel flame retardant DOPO-HQ has received considerable attention and exhibits prominent fire-resistant properties in epoxy resins and polyurethane foam [27]. DOPO-HQ possesses high thermal properties, chemical endurance and water resistance because of its rigid aromatic structures and stable P–O–C bond. On the other hand, DOPO-HQ can degrade to produce phosphorous-containing radicals which further prevent polymers from burning by scavenging H∙ and OH∙ radicals in gas phase [28].

In this paper, a novel functionalized flame retardant, HABP-DOPO, was designed and synthesized by substitution reaction based on cyclotriphosphazene, dihydroxybiphenyl and DOPO. HABP-DOPO was incorporated into the PLA matrix to prepare fire-resistant PLA/HABP-DOPO blends via a melt blending method. The chemical structure of HABP-DOPO was characterized by nuclear magnetic resonance (NMR), Fourier transform infrared spectroscopy (FTIR) and elemental analysis (EA). The thermal properties and fire-resistant performance of PLA/HABP-DOPO blends were conducted by TGA, DSC, LOI, cone colorimetry and UL-94 vertical combustion tests. The flame retardant mechanism and mechanical properties of blends were also investigated.

## 2. Materials and Methods 

### 2.1. Materials

PLA (4032D, 0.3 g/10 min) was purchased from Nature Works LLC (Minnetonka, MN, USA), the flame retardant HABP-DOPO was synthesized and hexachlorocyclotriphosphazene (HCCP), p-hydroxybenzaldehyde, 2,2′-biphenyldiol and DMF were purchased from Shanghai Aladdin Bio-Chem Technology Co., Ltd. (Shanghai, China).

### 2.2. Synthesis of the Flame Retardant HABP-DOPO

The synthetic route is illustrated in Figure 1. A mixture solution of 29 mmol HCCP, 29 mmol 2,2′-dihydroxybiphenyl and 70 mmol K_2_CO_3_ (Aladdin, Shanghai, China) in 100 mL acetone was stirred at room temperature for 15 min. The filtrate was steamed in a vacuum, then the residue was extracted with CH_2_Cl_2_ (Aladdin, Shanghai, China). Recrystallization from CH_2_Cl_2_/petroleum ether (Aladdin, Shanghai, China) gave the pure product C_12_H_8_O_2_C_l4_N_3_P_3_ [29]. FTIR: 1438.46, 1477.09, 1502.04 (benzene ring), 1176.44, 1209.64 (P=N) and 756.40 cm^−1^ (1, 2 ortho substitution on the benzene ring). ^1^H NMR (500 MHz, CDCl_3_): 7.32–7.60 (8 H, C_12_H_8_). ^31^P NMR: (CDCl_3_) 13.3 (1P, P(O_2_C_12_H_8_)) and 25.2 (2P, P–Cl). EA for C_12_H_8_O_2_C_l4_N_3_P_3_: C, 31.3; H, 1.7 and N, 9.1. 

In 150 mL tetrahydrofuran (THF) (Aladdin, Shanghai, China), 4.26 mmol C_12_H_8_O_2_C_l4_N_3_P_3_ was dissolved and slowly added to a mixture solution of P-hydroxybenzaldehyde and K_2_CO_3_. The solution was stirred for half an hour at room temperature and heated to 80 °C for 24 h to obtain intermediate HABP. FTIR: 1704.01 (the C=O of the –CHO), 2741.55, 2844.26 (the C–H of the –CHO), 1178.56 (the vibration peak of the aromatic aldehyde), 521.00 and 594.00 cm^−1^ (P–Cl) disappeared. ^1^H NMR (500 MHz, CDCl_3_) δ: 6.80–7.58 (8 H, C_12_H_8_) and 7.83–7.87 (16H, C6H4). ^31^PNMR: (CDCl_3_), 7.77, 8.35 (2P, P(O_2_C_7_H_5_)) and 23.40–24.57 (1P, P(O_2_C_12_H_8_)). EA for HABP: C, 59.6; H, 3.5 and N, 5.2.

In a 500 mL three-necked flask equipped with a magnetic stirrer, a condenser and a nitrogen apparatus at 120 °C, 26.4 mmol DOPO was dissolved in 100 mL *N*,*N*-Dimethylformamide (DMF). To the mixture, 4 mmol HABP was added and maintained stirred at 140 °C for 10 h. Finally, a large amount of ice water was fed into the solution and white precipitate was obtained. The precipitate was washed three times with dry ethanol/toluene mixed solution and dried in a vacuum for 24 h at 80 °C. FTIR: 3424.54 (–OH), 756.24, 946.96 (P–O–Ph), 1209.64 (P=N) and 2437.55 cm^−1^ (P–H) disappeared. ^1^H NMR (500 MHz, DMSO): 5.19–5.40 (H, C–H), 6.46–6.94 (H, –OH) and 7.01–8.04 (H, DOPO). ^31^P NMR (DMSO): 8.83, 9.40 (1P, P(C_12_H_8_)), 25.06 and 31.50 (2P, P(DOPO)). EA for HABP-DOPO: C, 12.7; H, 4.3 and N, 2.5.

### 2.3. Preparation of Flame Retardant PLA Blends

PLA was dried in a TY-ZK-1 vacuum (Taiyu, Suzhou, China) at 80 °C for 12 h. According to the mixing ratio, PLA and HABP-DOPO were blended in a melt mixer (Shanghai S.R.D. LB-100, China) at 180 °C and 60 rpm for 10 min. The mixture was hot-pressed into a sheet specimen under 5 MPa at 190 °C.

### 2.4. Measurements

FTIR spectra were recorded using a Thermo Nicolet iS50 spectrometer (Thermo Fisher Scientific inc.; Waltham, MA, USA) in the range of 500–4000 cm^−1^ (KBr disk). ^1^H NMR and ^31^P NMR data were obtained using a Bruker 400 MHz WB Solid-State NMR Spectrometer (Bruker, Billerica, MA, USA). Elemental analysis data were obtained using an instrument of Vario EL cube (Elementar, Hamburg, Germany). TGA was carried out on an STA 449 F3 Jupiter analyzer (Netzsch GmbH, Germany) under N_2_ atmosphere at a heating rate of 10 °C/min from room temperature to 800 °C. Thermal behavior of the samples was studied with a NETZSCH-DSC 200F3 (Netzsch GmbH, Germany) instrument under nitrogen atmosphere. Samples were firstly heated from 30 to 200 °C at a rate of 10 °C/min, held at 200 °C for 5 min, cooled to 30 °C at the cooling rate of 10 °C/min and heated again to 200 °C at the same rate.

LOI values were performed on a HC-2 (Jiangning Analysis Instrument Co., Nanjing, China) with 100 × 10 × 3 mm sheets according to ASTM D2863-97. A UL-94 burning test was carried out on a ZY6017 instrument (Zonsky Instrument CO., Ltd., Dongguan, China) with 125 × 13 × 4 mm sheets according to the UL-94 standard method. The fire test was performed on an FTT cone calorimeter (FTT Ltd., East Grinstead, UK) with 100 × 100 × 3 mm sheets at a heat flux of 50 kW/m^2^ in the horizontal configuration according to ISO 5660. The morphologies of the char after a cone calorimeter test was observed using JEOL JSM–7500F field emission SEM (JEOL, Tokyo, Japan). The surfaces were sputtered with a thin gold layer before examination. The notched Izod impact tests were performed with 80 × 10 × 4 mm specimens via a Shenzhen Suns PTM1251-B impact tester (Suns, Shenzhen, China) according to GB/T 1843-2008. The notched impact strength (*a_iN_*) of the samples was calculated by Equation (1). *E_c_* is the corrected samples fracture absorption energy, *h* is the thickness of samples and *b_N_* is the remaining width of the samples.
(1)$$a_{iN}=\frac{E_{c}}{h*b_{N}}*10^{3}$$

The tensile tests were measured using a Shenzhen Suns UTM6104 electronic universal tester (Suns, Shenzhen, China) with 5 mm/min in accordance with GB/T l040.3-2006. Tensile strength and elongation at break were performed by Equations (2) and (3). Σ is the tensile strength, *P* is the breaking load, and *b* and *d* are the width and thickness of samples. *Ε* is the elongation at break. *G*_0_ is the original gauge length of specimens and *G* is the mark line distance at break. At least seven specimens were tested, and the average values were reported.
(2)$$\Sigma =\frac{P}{bd}$$
(3)$$Ε=\frac{G-G_{0}}{G_{0}}\times 100\%$$

## 3. Results and Discussion

### 3.1. Limiting Oxygen Index and Vertical Burning Rating Test

Flame retardant properties of PLA/HABP-DOPO blends were investigated through LOI and UL-94 tests. The results of LOI, UL-94 rating and dripping behavior of PLA blends are summarized in Table 1. It can be seen that pure PLA has a low LOI and exhibits high combustion and heavy dripping. LOI values markedly increased to 22.1, 25.1 and 28.5 with addition of 15, 20 and 25 wt% HABP-DOPO. The increased LOI values show that burning behavior is restrained during flammation in higher oxygen conditions. HABP-DOPO promotes formation of a wrapped carbon layer during the combustion process, suppresses dripping of the PLA matrix and inhibits transfer of heat and oxygen into the inner substrate, which is the main reason for the higher LOI value [30]. Addition of HABP-DOPO can also decrease burning time significantly. Neat PLA cannot self-extinguish due to the low melt strength of PLA resin during combustion, while the UL-94 rating of the PLA blend with 25 wt% HABP-DOPO reaches V-0 level, and dripping behavior is clearly inhibited. With addition of HABP-DOPO, dripping of blends was inhibited. There was no dripping when the amount of HABP-DOPO was 25%. Results demonstrate that HABP-DOPO not only has a good effect on the dripping behaviors of PLA blends during combustion but also leads to enhancement of LOI values at higher flame retardant concentrations.

### 3.2. Thermogravimetric Analysis

Thermal stability of HABP-DOPO, neat PLA and its blends under nitrogen atmosphere are characterized by TGA and derivative thermogravimetry (DTG) in Figure 2. The data obtained from TGA and DTG curves for PLA and its blends are illustrated in Table 2. It can be seen that HABP-DOPO shows very high thermal stability. The initial decomposition temperature of HABP-DOPO is about 326.7 °C (based on 5% weight loss). According to DTG curves, the main peak of thermal degradation is about 476.8 °C, which can be attributed to the bisphenol structure and the P=N six-membered ring. Based on the TGA curve of the PLA blend with 25 wt% HABP-DOPO, its residue char at 500, 600, 700 and 800 °C is 17.4, 15.4, 15.0 and 14.4 wt%, respectively, while that of the neat PLA is 0.026 wt% at 400 °C. These results indicate that the PLA/HABP-DOPO blend presents higher thermal stability and has more char over 360 °C.

It can be observed from DTG curves that PLA displays one-step degradation behavior, and PLA/HDBP-DOPO blends exhibit two-steps processes. With the content of flame retardant increasing, the maximum decomposition temperature (*T_max_*) of the blends decreases gradually and the residue increases. This indicates that addition of HABP-DOPO has an obvious effect on thermal stability and degradation behaviors of PLA. The onset degradation temperature (*T_5%_*) of PLA/HABP-DOPO blends is lower than that of neat PLA due to the relatively low decomposition temperature of HABP-DOPO. Acid products of decomposed phosphaphenanthrene in HABP-DOPO promoted early degradation of PLA at the lower temperature. The lower appropriate decomposing temperature and pyrolysis products with quenching effect of HABP-DOPO all facilitated the flame retardant effect of HABP-DOPO on PLA. Moreover, as the temperature increased, the protective layer constructed by the sample in the early stage of heating also played a role in delaying the degradation process of the PLA sample [30,31].

### 3.3. Differential Scanning Calorimetry Test

Differential scanning calorimetry (DSC) curves of PLA and its blends are depicted in Figure 3, and detailed results of DSC are summarized in Table 3. It can be seen from Table 3 that the glass transition temperature (*T_g_*) of PLA/HABP-DOPO blends increased gradually with the content of HABP-DOPO increasing. As for the blend with 25 wt% HABP-DOPO, *T_g_* of the PLA blends has increased from 59.9 to 62.6 °C. An independent *T_g_* can be observed in DSC curves of all PLA/HABP-DOPO blends in Figure 3, indicating that there is no interface slippage between flame retardant and PLA matrix [32].

Addition of HABP-DOPO can improve the cold crystallization temperature (*T_cc_*) of PLA and weaken its non-isothermal crystallization ability. The reason for the disappearance of *T*_cc_ is that the induction effect of the flame retardant accelerates crystallization rate and the crystal formation is more complete. For PLA/25%HABP-DOPO blends, formation of aggregated particles in the polymeric matrix during melt processing reduces the nucleation point and makes crystallization incomplete: that is why the *T_cc_* peak reappears. Because HABP-DOPO as a dispersed phase affects the integrity of the PLA crystal and prevents regular accumulation of molecular chains, crystallinity (*X_c_*) of PLA decreases. With increase of HABP-DOPO content in the blends, the main melting peak (*T*_*m*2_) gradually weakened. Gradual addition of HABP-DOPO will reduce the ability of the material to form a crystal structure, resulting in a decrease in crystallinity of the composite material, which is manifested as a decrease in heat of crystallization (Δ*H_c_*) and heat of melting (Δ*H_m_*) of the composite material.

### 3.4. Cone Calorimetry Test 

In order to evaluate the fire hazard under real fire conditions, cone calorimeter tests are conducted to investigate the flame retardant and burning behaviors of PLA/HABP-DOPO blends. Total heat release (THR), heat release rate (HRR) and total smoke release (TSR) curves of PLA and its blends are presented in Figure 4, Figure 5 and Figure 6. Partial characteristic parameters, such as the time to ignition (TTI), peak of heat release rate (pk-HRR), average value of heat release rate (av-HRR), total heat release (THR), effective heat of combustion (EHC) and average value of mass loss rate (av-MLR) are summarized in Table 4. Digital photos of the blends after the cone calorimetry test are shown in Figure 7.

THR refers to total heat release of the materials released from ignition to flame extinction under preincident heat flow intensity. The THR value of neat PLA is 72.15 MJ/m^2^ and decreased with addition of HABP-DOPO. When the HABP-DOPO content was 15, 20 and 25 wt%, the THR values of PLA/HABP-DOPO blends were 60.85, 59.43 and 43.07 MJ/m^2^, respectively. The slopes of THR curves represent the flame diffusion rate of the material [33]. With the content of HABP-DOPO increasing, the slopes of PLA/HABP-DOPO blends gradually decrease and combustion of the flame has begun to slow down. The slope of PLA blend with 25 wt% HABP-DOPO shows the lowest value and exhibits good fire resistance.

HRR refers to heat release rate per unit area after a material is ignited under a preset incident heat flux. As shown in Table 4, when the content of HABP-DOPO is increased from 0 to 25 wt%, the pk-HRR value is reduced to 271.38 kW/m^2^ from 336.86 kW/m^2^.

Figure 5 shows HRR curves of PLA and its blends. Neat PLA had only one peak during the combustion process, while PLA/HDBP-DOPO blends presents two peaks. HRR values corresponding to two peaks are obviously lower than that of one peak of neat PLA. The first peak appears at 80–100 s, which can be attributed to rapid decomposition of HABP-DOPO on the surface which can cause degradation of the PLA matrix, which in turn induces the surface temperature of the blends to rise rapidly and generates heat released through combustion. The second peak appears around 150 s corresponding to burning of PLA/HABP-DOPO blends.

TTI of PLA/HABP-DOPO blends become short gradually with the content of HABP-DOPO increasing because of quick decomposition of HABP-DOPO. Effective heat of combustion (EHC) can evaluate the contribution of effective combustion components to heat release that exist in the gas phase during the combustion process [34]. As the content of HABP-DOPO increases, av-EHC decreases slightly, which indicates that the content of effective combustion components in the product was reduced.

Total smoke release (TSR) refers to the total cumulative amount of smoke generated when a unit sample area of a material is burned. As shown in Figure 6, with the increase of HABP-DOPO, TSR increases gradually, demonstrating that HABP-DOPO plays an important role in gas phase flame retardancy. Elevated TSR can reduce concentration of oxygen by diluting concentration of oxygen and taking away heat generated during the combustion process [35].

The residual char after the cone calorimetry test can intuitively show the flame retardant performance of polymer blends. Figure 7 shows photos of PLA/HABP-DOPO blends after the cone calorimetry test. The residual char increases with the content of HABP-DOPO increasing. This can be attributed to the following reasons. HABP-DOPO plays the role of a carbon-forming agent, and good dispersion of flame retardant in the PLA matrix also provides more carbon formation sites. In conclusion, addition of HABP-DOPO is beneficial to formation of a carbon layer in the solid phase.

### 3.5. Morphologies of the Residue Char

It is well known that morphologies of the char layer play a very important role in the performance of flame retardancy, profiting from prevention of heat transfer, flame spreading and droplet generation. In order to clarify the relationship between flame retardant performance and the microstructure of intumescing chars, the intumescing char residue of PLA/HABP-DOPO blends after cone calorimeter tests were examined using SEM as shown in Figure 8.

In Figure 8a,b for the PLA blend with 15 wt% HABP-DOPO, the surface of the char layer has many big holes and several cracks. The char layer reveals slight foaming, but the foams are not adequately dense due to insufficient char formation or less condensed char during the burning process. The exchange channels, originated from heat and gas generated during the combustion process, are not conducive to flame retardancy. These factors determine poor flame retardancy performance of this blend.

However, the residue of the PLA blend with 25 wt% HABP-DOPO exhibits denser foams with intumescent and compact morphologies. This structure of char can hinder both mass and heat transfer to effectively retard degradation of the PLA matrix and block gas and heat exchange during the combustion process. Results indicate that the carbon layer formed by 25 wt% HABP-DOPO in the solid phase is more stable and can withstand the impact of high temperature radiation and gas release [30]. A large amount of HABP-DOPO plays a role in solid phase flame retardancy. It also can be observed from Figure 8 that flame retardant particles are presented on the surface of the residual char. This is because the flame retardant gradually migrates, deposits on the surface of the material and forms a wrinkled residual char during combustion of the PLA blends. Flame retardants promote formation of the residual char.

### 3.6. FTIR of PLA/HABP-DOPO Blends

Infrared spectroscopy is used to characterize the existence of hydrogen bonds in polymer systems. The C=O groups of PLA are generally considered to be a hydrogen bonding acceptor, while the OH groups of HABP-DOPO are considered to be a hydrogen-bonded donor. Figure 9 shows FTIR spectra of neat PLA and its blend. Compared with neat PLA, the peaks of PLA/HABP-DOPO blend have changed significantly. A large number of –OH groups of HABP-DOPO can produce strong intermolecular hydrogen bonds: O–H^...^O with the O atoms in PLA, thereby improving their compatibility. Due to in-plane bending, the COOH peak of pure PLA at 1360.7 cm^−1^ is split into 1360.7 and 1382.1 cm^−1^ bands. The narrow symmetrical peak (at 1757 cm^−1^), corresponding to the C=O stretching mode of the ester in PLA, becomes wider and eventually splits into two bands. It is attributed to addition of HABP-DOPO. The amide peak shows obvious weak splitting and a new weak peak gradually forms at the low wave number of the amide peak [36].

### 3.7. Dispersion of Flame Retardant in PLA Matrix

Morphologies and distributions of the additive-type flame retardant in PLA play an important role in mechanical properties as well as flame retardancy. The SEM micrograph of the cryo-fracture of PLA/HABP-DOPO blends is displayed in Figure 10. It is obvious that HABP-DOPO particle agglomerations cannot be observed in SEM micrographs with different magnifications, which meant a uniform distribution of the flame retardant particles in the PLA matrix [22,37]. 

### 3.8. Mechanical Properties Test

Usually, introduction of flame retardants to the polymer can lead to decline of mechanical properties because of their poor compatibility. Figure 11 presents impact properties, tensile properties and elongation at break properties of neat PLA and PLA/HABP-DOPO blends. The Izod impact strength of PLA was 4.51 kJ/m^2^ and its blends are 15, 20 and 25 wt%. For HABP-DOPO the Izod impact strengths were 4.37, 4.29 and 4.21 kJ/m^2^, respectively. Compared to pure PLA, addition of HABP-DOPO has little effect on the impact properties of PLA, which could be attributed to effective dispersion of flame retardants. Hydrogen bonding between the hydroxyl-rich flame retardant and the PLA matrix is another important reason for good compatibility. Similar to Izod impact strength, tensile strength and elongation at break of the blends change little with the content of HABP-DOPO increasing. Therefore, HABP-DOPO is a powerful candidate for PLA because PLA/HABP-DOPO blends exhibit balanced mechanical properties and flame retardancy.

### 3.9. Mechanism of Flame Retardant in PLA Matrix

By comparing effective heat of combustion (EHC) of neat PLA and its blends, we can evaluate whether the flame retardant mechanism is mainly in the gas phase or solid phase. If the system is a gas phase flame retardant mechanism, the flame retardant burned will release free radicals in the gas phase. These free radicals will stop the combustion reaction, PLA blends will not completely burn and the EHC of PLA blends will be lower than that of neat PLA. Similarly, if the EHC of PLA blends is higher than that of the PLA matrix, the flame retardants exhibit the solid phase flame retardant mechanism. 

Figure 12 shows EHC curves of PLA and PLA/HABP-DOPO blends. It can be seen that in the first stage of combustion, the solid phase is mainly flame retardant and the DOPO in HABP-DOPO is a carbon-forming agent at a high temperature stage. The reason is that stability of the phosphazene structure is higher than that of the phosphaphenanthrene structure. Based on analysis of microscopic morphology after cone calorimetry, a large amount of residual carbon forms during the flame retardant process in the condensed phase and can act as heat insulation and an oxygen barrier to protect the matrix. It can be seen that most of it presents a relatively complete, dense and thick carbon layer structure, and there are also relatively fluffy, porous or even expanded morphologies, which are mainly caused by emission of flame retardant gases such as ammonia during the combustion process. In the second stage of combustion, HABP-DOPO is thermally decomposed to form a series of radicals containing P such as Ph–O and HPO_2_, which can play a role in promoting dehydration and accelerating char formation. At the same time, non-combustible gases such as NH_3_, H_2_O and CO_2_ will dilute O_2_ and gaseous fuel concentration in the combustion zone and exhibit good flame retardancy. As shown in Figure 13, the flame retardant mechanism of PLA/HABP-DOPO blends should be the synergistic flame retardancy of solid phase and gas phase [38,39].

## 4. Conclusions

A novel flame retardant (HABP-DOPO) containing phosphazene and phosphaphenanthrene was synthesized and blended with PLA to improve its flame retardancy. The PLA blend with 25 wt% HABP-DOPO had a 28.5% LOI value and UL-94 V-0 flammability rating. The pk-HRR, THR and av-MLR values of PLA/HABP-DOPO blends obtained from CCT reduced dramatically compared to pure PLA. The onset degradation temperature of PLA/HABP-DOPO blends decreased with the content of HABP-DOPO increasing. The morphologies of the char residues had a continuous, compact and intumescent outer surface, hindered heat transmission and gas diffusion, and exhibited good flame retardancy. Furthermore, PLA/HABP-DOPO blends maintain the same level of mechanical properties as neat PLA. Therefore, HABP-DOPO is a powerful flame retardant. The flame retardant mechanism of PLA/HABP-DOPO blends should be the synergistic flame retardancy of solid and gas phases.

## Figures and Tables

**Figure 1 polymers-12-02407-f001:**
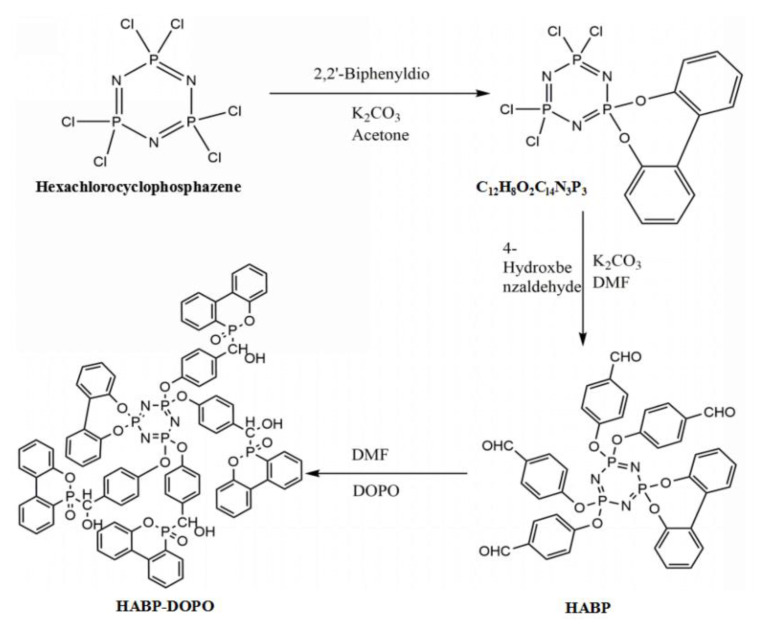
Preparation route and structure of HABP-DOPO.

**Figure 2 polymers-12-02407-f002:**
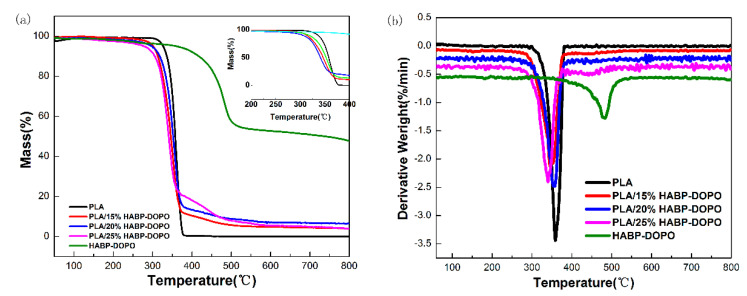
TGA and DTG curves of HABP-DOPO, neat PLA and PLA/HABP-DOPO blends under nitrogen gas atmosphere. (**a**) TGA curves and (**b**) DTG curves.

**Figure 3 polymers-12-02407-f003:**
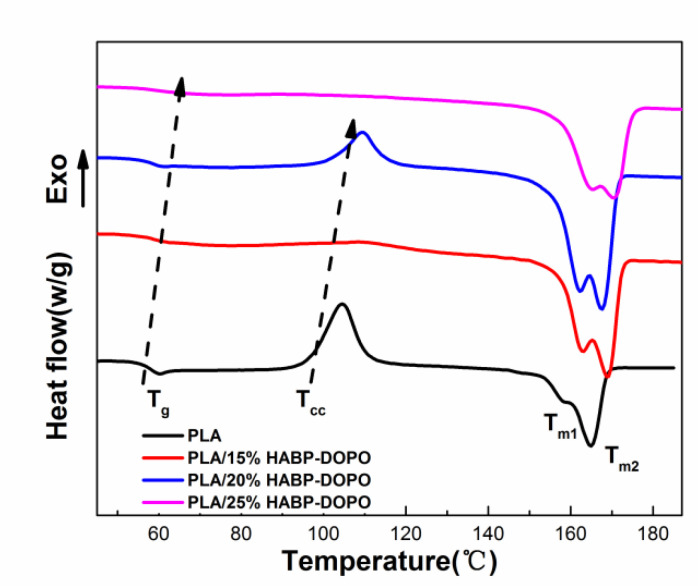
DSC curves of PLA and its blends.

**Figure 4 polymers-12-02407-f004:**
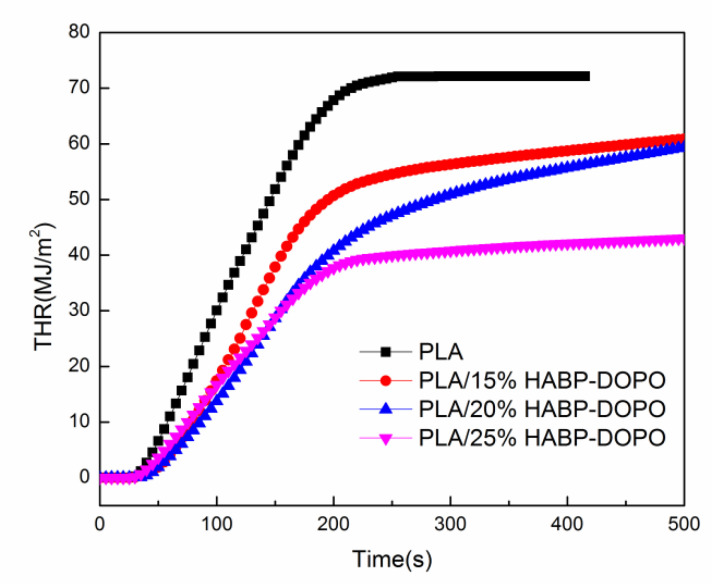
Total heat release (THR) curves of PLA and PLA/HABP-DOPO blends.

**Figure 5 polymers-12-02407-f005:**
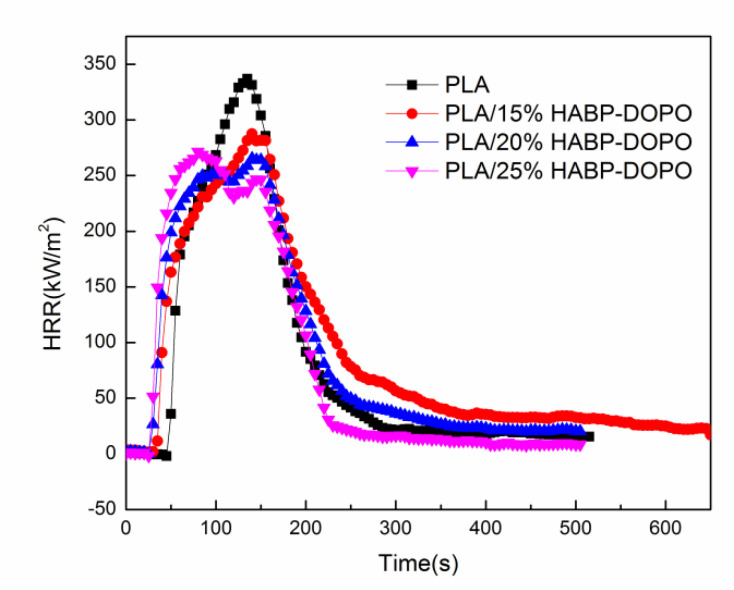
Heat release rate (HRR) curves of PLA and PLA/HABP-DOPO blends.

**Figure 6 polymers-12-02407-f006:**
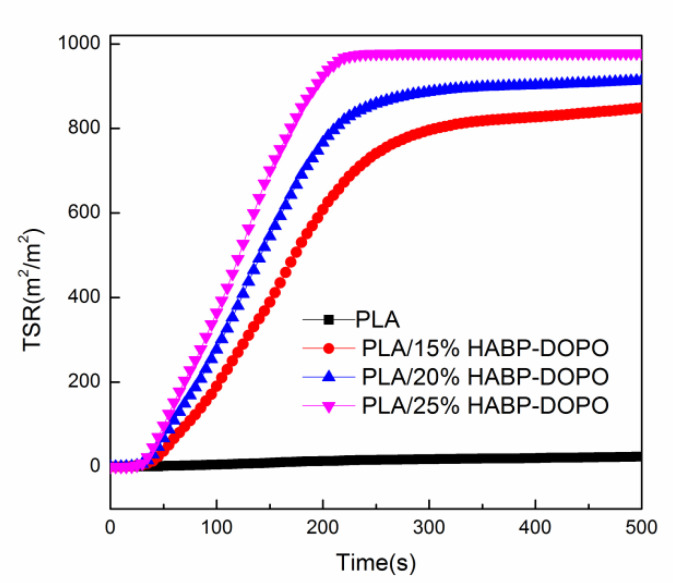
Total smoke release (TSR) curves of PLA and PLA/HABP-DOPO blends.

**Figure 7 polymers-12-02407-f007:**
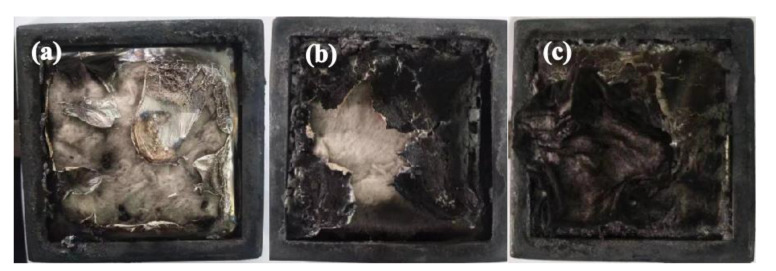
Digital photos of PLA/HABP-DOPO blends after cone calorimetry test: (**a**) pure PLA, (**b**) PLA/15%HABP-DOPO and (**c**) PLA/25%HABP-DOPO.

**Figure 8 polymers-12-02407-f008:**
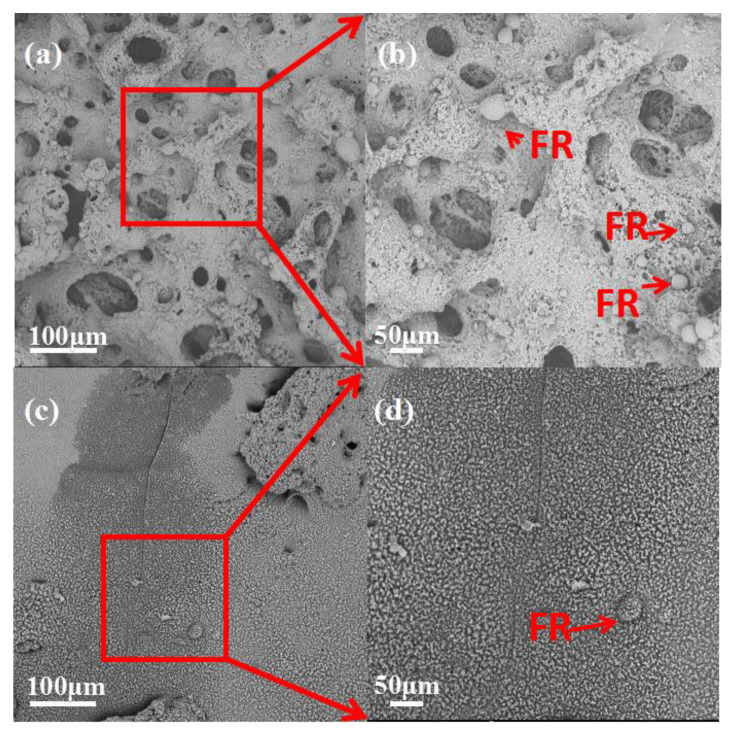
SEM images of PLA/HABP-DOPO blends after cone calorimeter tests. (**a**), (**b**) PLA/15 wt% HABP-DOPO and (**c**), (**d**) PLA/25 wt% HABP-DOPO.

**Figure 9 polymers-12-02407-f009:**
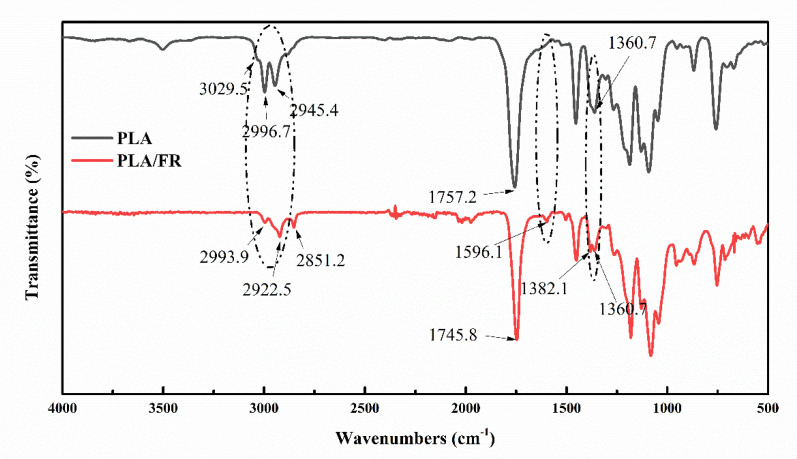
FTIR of neat PLA and its blend with 25 wt% HABP-DOPO.

**Figure 10 polymers-12-02407-f010:**
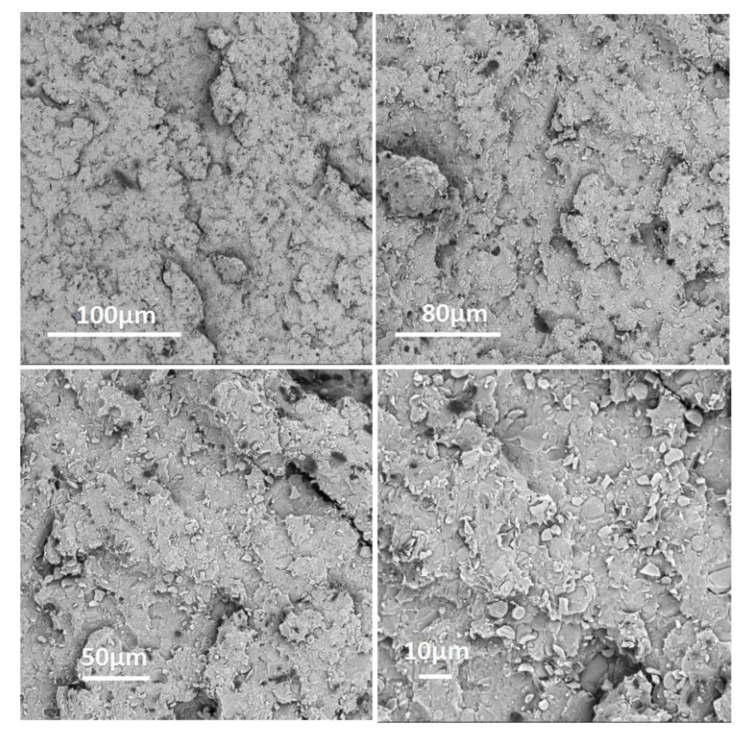
SEM micrograph of the cryo-fracture of the PLA/25% HABP-DOPO composite.

**Figure 11 polymers-12-02407-f011:**
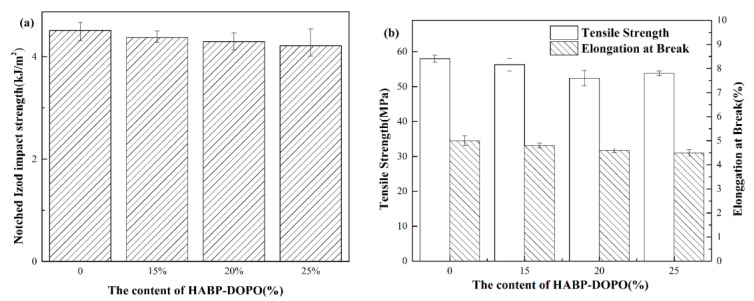
Mechanical properties of PLA and its blends: (**a**) notched impact strength and (**b**) tensile strength and elongation at break.

**Figure 12 polymers-12-02407-f012:**
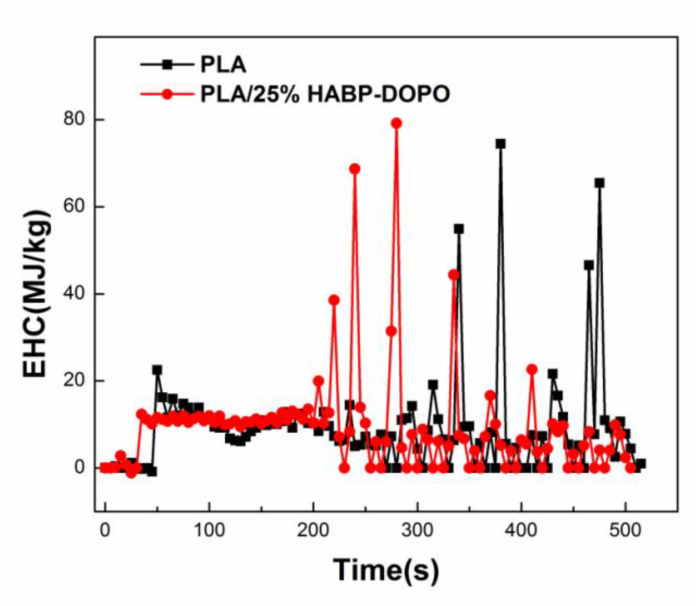
Effective heat of combustion (EHC) curves of PLA and PLA/HABP-DOPO blends.

**Figure 13 polymers-12-02407-f013:**
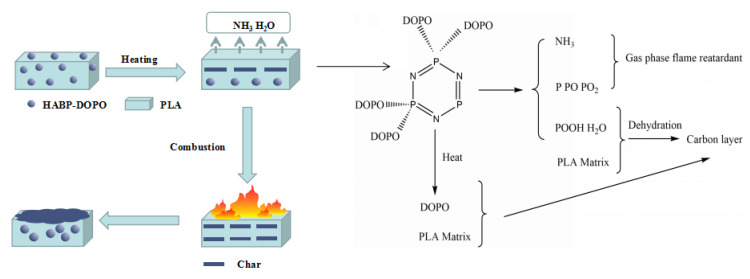
Flame retardant mechanism of PLA/HABP-DOPO blends.

**Table 1 polymers-12-02407-t001:** The limiting oxygen index (LOI) values and vertical burning (UL-94) rating of PLA and its blends.

	LOI (%)	UL-94	Dripping	Ignition
PLA	18.5	NR	Y	Y
PLA/15 wt% HABP-DOPO	22.1	V-1	Y	Y
PLA/20 wt% HABP-DOPO	25.1	V-1	Y	Y
PLA/25 wt% HABP-DOPO	28.5	V-0	N	N

**Table 2 polymers-12-02407-t002:** Thermal properties of HABP-DOPO, neat PLA and PLA/HABP-DOPO blends.

	*T_5%_* (°C)	*T_max_ (*°C)	*T_25%_* (°C)	Residue (wt%) at 800 °C
PLA	326.7	357.2	348.2	0
HABP-DOPO	359.4	476.8	470.3	47.7
PLA/15 wt% HABP-DOPO	301.4	344.5	330.3	3.50
PLA/20 wt% HABP-DOPO	297.5	352.3	332.0	8.68
PLA/25 wt% HABP-DOPO	307.6	339.2	337.9	14.43

**Table 3 polymers-12-02407-t003:** Detailed results of DSC of PLA and its blends.

Sample	*T_g_*(°C)	*T_cc_*(°C)	Δ*H_c_*(J/g)	*T*_*m*1_(°C)	*T*_*m*2_(°C)	*ΔH*_*m*1_(J/g)	*X_c_*(%)
PLA	59.9	107.7	35.46	150.1	168.2	49.91	53.3
PLA/FR15	62.6	109.8	1.583	163.2	168.7	29.34	36.8
PLA/FR20	60.9	109.5	8.907	162.7	167.2	35.49	47.3
PLA/FR25	62.6	-	1.668	165.6	170.6	26.17	37.2

**Table 4 polymers-12-02407-t004:** Cone calorimetric test data for PLA/HABP-DOPO blends.

Sample	TTI (s)	Pk-HRR (kW/m^2^)	Av-HRR (kW/m^2^)	THR (MJ/m^2^)	TSR (m^2^/m^2^)	Av-MLR g/(s·m²)	EHC (MJ/kg)
PLA	53	336.86	263.42	72.15	22.2	20.58	74.47
PLA/15 wt% HABP-DOPO	39	322.9	241.35	63.7	860.6	12.64	70.14
PLA/20 wt% HABP-DOPO	36	303.2	231.7	53.8	909.9	18.78	73.36
PLA/25 wt% HABP-DOPO	32	271.38	212.54	42.9	970.2	25.87	79.15

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
