# Peer review of "Effect of a Novel Flame Retardant on the Mechanical, Thermal and Combustion Properties of Poly(Lactic Acid)"

_polymers, 2020, doi:10.3390/polym12102407_

Round 1
Reviewer 1 Report
Summary
The authors present a new flame retardant additive “HABP-DOPO” which can be used for applications in PLA polymers. The study shows sufficient flame retardant properties (V-0; LOI: 28,5%) for PLA containing 25wt.% HABP-DOPO without a significant loss in mechanical properties. Good flame retardant properties are confirmed by Cone Calorimeter tests where most key figures improve with higher HABP-DOPO contents.
The study is mainly limited to phenomenological analytics. More insights of how the new flame retardant works in PLA would be appreciated.
General comments:
- English should be improved; check for comma errors and objective language
- Please improve all figures graphic quality
Specific comments:
Line 35: improvement of the line arrangement is suggested
Line 67-70: Please remove the enumeration (1-3)
Line 67: Please check company reference; suggestion NatureWorks LLC (Minnetonka, Minnesota, USA)
Line 80+87: THF and DMF have not yet been introduced; please add long name
Line 100: Please check company reference; suggestion Thermo Nicolet IS50 spectrometer (Thermo Fisher Scientific inc.; Waltham, Massachusetts, USA)
Line 104: Please check company reference; suggestion (STA 449 F3 Jupiter, Netzsch GmbH, Germany)
Line 106: Please check company reference; suggestion (Netzsch GmbH, Germany)
Line 112: Please check company reference; suggestion Fire Testing Technology ltd. (East Grinstead, UK)
Line 115: Please check company reference; suggestion JEOL JSM-7500F field emission SEM (Tokio, Japan)
Line 116: Please check company reference; suggestion PTM1251-B impact tester (Company, City, China)
Line 118: Please check company reference; suggestion Shenzhen Suns UTM6104 electronic universal tester (Company, City, China)
Line 126-128: Rephrase word repetition “during”
Line 129: please specify “outside world”
Line 134: please specify “good effect”
Line 165: Table 3; please define Tcc
Line 193: please rephrase “heat sum”
Line 201: table 4; THR Value for pure PLA should be 72.15MJ/m²;
Line 336: flame retardants à flame retardant
Figure 3: please subscribe why the Tcc of PLA containing 15% and 25% is not visible and for 0% and 15% is visible
Author Response
We wish to extend our deepest appreciation to you for valuable suggestions about our manuscript titled "Effect of a Novel Flame Retardant on the Mechanical, Thermal and Combustion Properties of Poly (Lactic Acid)" (POLYMERS-946141). Based on the valuable comments and criticisms raised by you, we revised our original paper very carefully and answered all the comments and criticisms one by one.
Comment-1. The study is mainly limited to phenomenological analytics. More insights of how the new flame retardant works in PLA would be appreciated.
Response-1: Thank you very much for this valuable suggestion. We tried to supplement the flame retardant mechanism. The role of flame retardants in the gas phase and the solidified phase is more supplemented. The presence of porous and complete residual layers proves that the flame retardant works in both the solid phase and the gas phase. Ph-O and HPO2 produced by the decomposition of the phosphazene ring promote the carbonization of the PLA matrix.
Comment-2. English should be improved; check for comma errors and objective language. Please improve all figures graphic quality.
Response-2: Thank you very much for this valuable suggestion. We found some grammatical errors and made corrections. The image quality has been upgraded to 300×300dpi.
Comment-3. Line 35: improvement of the line arrangement is suggested.
Response-3: Thank you very much for this valuable suggestion. We have made corresponding modification to the Line 35, and the corresponding revision has been done in the revised manuscript.
Comment-4. Line 67-70: Please remove the enumeration (1-3).
Response-4: Thank you very much for this valuable suggestion. Enumeration (1-3) have been removed, and the corresponding revision has been done in the revised manuscript.
Comment-5. Line 67: Please check company reference; suggestion NatureWorks LLC (Minnetonka, Minnesota, USA).
Response-5: Thank you very much for this valuable suggestion. The company reference has been modified to NatureWorks LLC (Minnetonka, Minnesota, USA), and the corresponding revision has been done in the revised manuscript.
Comment-6. Line 80+87: THF and DMF have not yet been introduced; please add long name.
Response-6: Thank you very much for this valuable suggestion. According to the reviewers’ comments, the full names of THF and DMF have been added.
Comment-7. Line 100: Please check company reference; suggestion Thermo Nicolet IS50 spectrometer (Thermo Fisher Scientific inc.; Waltham, Massachusetts, USA).
Response-7: Thank you very much for this valuable suggestion. According to the reviewers’ comments, the company reference has been updated.
Comment-8. Line 104: Please check company reference; suggestion (STA 449 F3 Jupiter, Netzsch GmbH, Germany).
Response-8: Thank you very much for this valuable suggestion. According to the reviewers’ comments, the company reference has been updated.
Comment-9. Line 106: Please check company reference; suggestion (Netzsch GmbH, Germany).
Response-9: Thank you very much for this valuable suggestion. According to the reviewers’ comments, the company reference has been updated.
Comment-10. Line 112: Please check company reference; suggestion Fire Testing Technology ltd. (East Grinstead, UK).
Response-10: Thank you very much for this valuable suggestion. According to the reviewers’ comments, the company reference has been updated.
Comment-11. Line 115: Please check company reference; suggestion JEOL JSM-7500F field emission SEM (Tokio, Japan).
Response-11: Thank you very much for this valuable suggestion. According to the reviewers’ comments, the company reference has been updated.
Comment-12. Line 116: Please check company reference; suggestion PTM1251-B impact tester (Company, City, China).
Response-12: Thank you very much for this valuable suggestion. According to the reviewers’ comments, the company reference has been updated.
Comment-13. Line 118: Please check company reference; suggestion Shenzhen Suns UTM6104 electronic universal tester (Company, City, China).
Response-13: Thank you very much for this valuable suggestion. According to the reviewers’ comments, the company reference has been updated.
Comment-14. Line 126-128: Rephrase word repetition “during”.
Response-14: Thank you very much for this valuable suggestion. According to the reviewers’ comments, “During the limit oxygen index test” has been removed.
Comment-15. Line 129: please specify “outside world”.
Response-15: Thank you very much for this valuable suggestion. According to the reviewers’ comments, the “outside world” has been described to “inhibits the transfer of heat and oxygen into the inner substrate”.
Comment-16. Line 134: please specify “good effect”.
Response-16: Thank you very much for this valuable suggestion. According to the reviewers’ comments, the “good effect” has been described to “With the addition of HABP-DOPO, the dripping of blends was inhibited. There was no dripping when the amount of HABP-DOPO was 25%”.
Comment-17. Line 165: Table 3; please define Tcc.
Response-17: Thank you very much for this valuable suggestion. According to the reviewers’ comments, the “Tcc” has been described to “cold crystallization temperature”.
Comment-18. Line 193: please rephrase “heat sum”.
Response-18: Thank you very much for this valuable suggestion. According to the reviewers’ comments, the “heat sum” has been exchanged to “total heat release”.
Comment-19. Line 201: table 4; THR Value for pure PLA should be 72.15MJ/m².
Response-19: Thank you very much for this valuable suggestion. According to the reviewers’ comments, the THR value of pure PLA has been updated.
Comment-20. Line 336: flame retardants à flame retardant.
Response-20: Thank you very much for this valuable suggestion. According to the reviewers’ comments, the “flame retardants” has been exchanged to “flame retardant”.
Comment-21. Figure 3: please subscribe why the Tcc of PLA containing 15% and 25% is not visible and for 0% and 15% is visible.
Response-21: Thank you very much for this valuable suggestion. According to the reviewers’ comments, we explained the reasons for the change in the Tcc curves. The reason for the disappearance of the Tcc is that the induction effect of the flame retardant accelerates the crystallization rate and the crystal formation is more complete. For PLA/25%HABP-DOPO blends, the formation of aggregated particles in the polymeric matrix during the melt processing reduces the nucleation point and makes the crystallization incomplete, that is why the Tcc peak reappears.

Reviewer 2 Report
The methods of mechanical tests as well as a number of tests for the average values calculation must be presented.
The choice of a maximum percentage (25%) of HABP-DOPO in the blends is not explained. Why did the authors do not investigated blends with a higher percentage of HABP-DOPO if all properties were increased? Why were used 15, 20 and 25% blends if the results of their properties are so close? Some optimisation of blends would increase scientific quality of paper.
Author Response
We wish to extend our deepest appreciation to you for valuable suggestions about our manuscript titled "Effect of a Novel Flame Retardant on the Mechanical, Thermal and Combustion Properties of Poly (Lactic Acid)" (POLYMERS-946141). Based on the valuable comments and criticisms raised by you, we revised our original paper very carefully and answered all the comments and criticisms one by one.
Comment-1.
The methods of mechanical tests as well as a number of tests for the average values calculation must be presented.
Response-1: Thank you very much for this valuable suggestion. According to the reviewers’ comments, we have presented some mechanical performance testing methods and the formulas used in their calculations.
Comment-2.
The choice of a maximum percentage (25%) of HABP-DOPO in the blends is not explained. Why did the authors do not investigated blends with a higher percentage of HABP-DOPO if all properties were increased? Why were used 15, 20 and 25% blends if the results of their properties are so close? Some optimisation of blends would increase scientific quality of paper.
Response-2: Thank you very much for this valuable suggestion. According to the reviewers’ comments, we explained why the maximum content of flame retardant is 25%. According to our experimental results, when the proportion of flame retardant is increased to more than 25%, compared with the blend with 25% content, the improvement of flame retardant performance is very limited, and the addition of more proportion of flame retardant is easy to cause agglomeration during processing, the mechanical properties decline seriously. The performance of blends with a content of 20-25% is better than those used in daily life, so we chose 25% as the maximum percentage in the blend.

Round 2
Reviewer 2 Report
Could be accepted